# The Occurrence of Five Unregulated Mycotoxins Most Important for Traditional Dry-Cured Meat Products

**DOI:** 10.3390/toxins14070476

**Published:** 2022-07-12

**Authors:** Tina Lešić, Ana Vulić, Nada Vahčić, Bojan Šarkanj, Brigita Hengl, Ivica Kos, Tomaž Polak, Nina Kudumija, Jelka Pleadin

**Affiliations:** 1Laboratory for Analytical Chemistry, Croatian Veterinary Institute, Savska Cesta 143, 10000 Zagreb, Croatia; lesic@veinst.hr (T.L.); vulic@veinst.hr (A.V.); kudumija@veinst.hr (N.K.); 2Laboratory for Quality Control in the Food Industry, Faculty of Food Technology and Biotechnology, University of Zagreb, Pierottijeva 6, 10000 Zagreb, Croatia; nvahcic@pbf.hr; 3Department of Food Technology, University North, Žarka Dolinara 1, 48000 Koprivnica, Croatia; bsarkanj@unin.hr; 4Department for Nutrition and Data Management, Center for Food Safety, Croatian Agency for Agriculture and Food, Ivana Gundulića 36b, 31000 Osijek, Croatia; brigita.hengl@hapih.hr; 5Department of Animal Science and Technology, Faculty of Agriculture, University of Zagreb, Svetošimunska Cesta 25, 10000 Zagreb, Croatia; ikos@agr.hr; 6Department of Food Science and Technology, Biotechnical Faculty, University of Ljubljana, Jamnikarjeva ulica 101, 1000 Ljubljana, Slovenia; tomaz.polak@bf.uni-lj.si

**Keywords:** sausages, hams, cyclopiazonic acid, aflatoxin B_1_, ochratoxin A, sterigmatocystin, citrinin, LC-MS/MS

## Abstract

This study investigated the occurrence of 5 unregulated mycotoxins in a total of 250 traditional dry-cured meat products sampled in 2020 and 2021 in five Croatian regions (eastern, northern, central, western, and southern). Aflatoxin B_1_ (AFB_1_), ochratoxin A (OTA), sterigmatocystin (STC), citrinin (CIT), and cyclopiazonic acid (CPA) concentrations were related to the geographical region of the product’s origin and to local weather. The results revealed the contamination of 27% of samples, namely, STC in 4% of samples in concentrations of up to 3.93 µg/kg, OTA in 10% of samples in concentrations of up to 4.81 µg/kg, and CPA in 13% of samples in concentrations of up to 335.5 µg/kg. No AFB_1_ or CIT contamination was seen. Although no statistically significant differences in concentrations of individual mycotoxins across the production regions were found, differences in mycotoxin occurrence were revealed. The eastern and western regions, with moderate climate, delivered the largest number of contaminated samples, while the southern region, often compared with subtropics, delivered the smallest, so that the determined mycotoxins were probably mainly produced by the *Penicillium* rather than the *Aspergillus* species. Due to the interaction of various factors that may affect mycotoxin biosynthesis during production, the detected concentrations cannot be related solely to the weather.

## 1. Introduction

Dry-cured meat products are traditional foods produced and consumed worldwide and are typical of the diets in European Mediterranean countries [1]. Croatian traditional dry-cured meat products encompass various dry-fermented sausages and dry-cured meats (prosciutto, pancetta, dry-neck, etc.) [2]. In the traditional production process, in addition to differences in recipes followed by the producing households, one also finds significant differences in hygienic and environmental production conditions that affect the specificity of microflora and yield differences in quality and safety of the final products [3,4]. One of the safety aspects that raises concern is contamination with toxigenic moulds and their secondary metabolites, mycotoxins [5].

The mycotoxin contamination of meat products can therefore be the direct consequence of moulds, mostly of the *Aspergillus* and the *Penicillium* genera, that overgrow their surface during the ripening process and play a role in the development of the product’s sensory properties [6,7,8]. Traditional production is characterized by uncontrolled key parameters such as temperature and relative humidity as well as the lack of microbiological filters and pressure barriers, enabling an intense and undesired mould growth that can consequently lead to unfavourable sensory properties and the mycotoxin contamination of traditional meat products (TMPs) [4,9]. In addition to direct mould contamination, mycotoxins can contaminate meat products due to natural but contaminated spices and other ingredients, or due to indirect transfer from farm animals exposed to naturally contaminated feed (carry-over effect) [10,11,12].

Investigation into the occurrence of mycotoxins in meat products is important for assessing human exposure to these contaminants through TMPs, which represent a significant part of some diets, as well as for assessing the risks for human health and establishing the maximum permitted levels of mycotoxins in meat products, which are not yet established in the European Union [13,14,15].

Due to their possible appearance in dry-cured meat products and their toxicity, the most interesting mycotoxins in this context are ochratoxin A (OTA) and aflatoxin B_1_ (AFB_1_), which were the focus of studies conducted in many countries [1,16,17,18]. OTA is a common contaminant in agricultural products such as cereals, coffee, dry fruit, spices, and wine, but it can also be found in animal meat products [19]. AFB_1_ can also be found in products of plant origin, such as tree nuts, peanuts, dried fruit, cereals, corn, and spices, but its presence is also not uncommon in products of animal origin such as milk and meat products. AFB_1_ has been identified as the most harmful oral carcinogen included by the International Agency for Research of Cancer (IARC) [17] in Group 1, while OTA with its nephrotoxic, genotoxic, neurotoxic, immunotoxic, teratogenic, and carcinogenic effects has been classified as a Group 2B carcinogen [16,19,20,21].

In addition to AFB_1_ and OTA, some mould species identified on the surfaces of these products can produce other mycotoxins, such as cyclopiazonic acid (CPA), citrinin (CIT), and sterigmatocystin (STC), but the knowledge on the incidence of these mycotoxins and consumer exposure to them is insufficient even though all of them have been shown to be toxic [14,22,23]. The toxic effects of STC are similar to those of AFB_1_, but this mycotoxin is less toxic and is listed as a Group 2B potential human carcinogen [23]. Due to the scarce toxicity data and the insufficiency of relevant carcinogenicity studies, the IARC has still not declared an acceptable CPA toxicity level, but it is known that it can damage the digestive organs, the myocardium, and the skeletal muscles [24]. Just like OTA, CIT can act as hepatotoxin, teratogen, foetotoxin, and genotoxin, but the IARC still tags it as a Group 3 compound that cannot be classified as a human carcinogen [25]. A few studies on CPA occurrence in meat products conducted so far have reported this occurrence to be high [26,27,28,29]. To the best of our knowledge, studies on STC occurrence in meat products have not been conducted yet, despite the EFSA CONTAM Panel recommendation to collect data on STC in food and feed using highly sensitive analytical methods with LOQ of less than 1.5 μg/kg. This type of research has been encouraged because the data on STC incidence are limited, making it impossible to assess human and animal exposure [14,30]. The occurrence of CIT was examined in only one study of Croatian meat products, despite the fact that the strains of one of the main mycotoxin producers *P. citrinum* are often isolated from dry-cured meat products [27,31].

The production of mycotoxins in meat products is influenced by various biological and environmental factors. The formation of mycotoxins can be climatically conditioned, so that their occurrence may vary across production years [32,33]. The aim of this study was to investigate the occurrence of five yet-unregulated but most important mycotoxins when it comes to traditional dry-cured meat products. In view of uncontrolled traditional production environments and already recognized weather impacts on the occurrence of toxicogenic moulds and mycotoxins, another aim of this study was to relate data on regional weather during the products’ maturation to mycotoxin concentrations determined in TMPs collected in five Croatian regions during two sampling years.

## 2. Results and Discussion

### 2.1. Method Validation

The results of the validation process are shown in Table 1. The calibration range for AFB_1_ was 0.05 to 1 ng/mL, 0.2 to 5 ng/mL for OTA, 0.1 to 10 ng/mL for STC, 1 to 10 ng/mL for CIT, and 0.5 to 25 ng/mL for CPA. The average recoveries ranged from 91% to 119%, i.e., fell into the required range of −50% to +20% [34]. The matrix effect was expressed as a slight ion enhancement in the 0.7–7.7% range (the lowest for CIT and the highest for STC). The lowest LOD and LOQ values were achieved with STC and AFB_1_ and the highest with CPA. For example, the multi-mycotoxin LC–MS/MS method of Dada et al. [35], used to analyse dried beef, which employs a solid–liquid extraction, resulted in the following LOD values: OTA 0.13 μg/kg, AFB_1_ 2 μg/kg, CIT 18 μg/kg, CPA 7 μg/kg, and STC 8 μg/kg. The validation results showed that the method is suitable for studying the occurrence of STC, CIT, OTA, AFB_1_, and CPA in dry-cured meat products. The validation results for OTA, AFB_1_, and CPA (LOD, LOQ, and recovery) were partly published earlier in the studies by Vulić et al. [26] and Kudumija et al. [36].

### 2.2. The Occurrence of Mycotoxins in Dry-Cured Meat Products

The investigated mycotoxins were detected in 27% of the analysed TMP samples, namely, STC in the concentrations from 0.10 to 3.93 µg/kg in 4% of the samples, OTA in the concentration range of 0.24–4.81 µg/kg in 10% of the samples and CPA in the concentrations of 2.25–335.5 µg/kg in 13% of the samples, while AFB_1_ and CIT were not detected in any of the TMP samples. CPA was determined in the highest percentage of samples in very high concentrations, followed by OTA.

The obtained statistical data on the detected mycotoxins (CPA, OTA, and STC) were summarized and presented in boxplots (Figure 1). The mean CPA concentration was 29.11 ± 59.31 µg/kg, with the median of 9.90 µg/kg. In the majority of samples, CPA concentrations were up to 45 µg/kg, except for the three extremes of 66.35 µg/kg, 108.5 µg/kg, and 335.5 µg/kg. The mean OTA concentration was 1.02 ± 1.09 µg/kg, with a median of 0.66 µg/kg. In most samples, the concentrations were up to 1.7 µg/kg, but they also showed three extremes: 2.66 µg kg, 3.02 µg/kg, and 4.81 µg/kg. The mean STC concentration was 0.58 ± 1.13 µg/kg, with a median of 0.16 µg/kg. The concentrations were mostly up to 0.70 µg/kg, except for a single extreme of 3.93 µg/kg.

The possible co-occurrence of CIT and OTA, AFB_1_ and STC, and AFB_1_ and CPA in different foods has also been described in the literature [14,23,24], but since the mycotoxins AFB_1_ and CIT were not detected at all, OTA and CPA co-occurrence was recorded in only 4 samples; OTA and STC were found to co-exist in only 2 samples but not as a consequence of common mould producers.

According to the current knowledge, research results on the incidence of STC in meat and meat products are not available. Its presence has been documented in feeds, cereals, bread, tree nuts, coffee grains, spices, beer, and cheese [14], and there is a study on its occurrence in animal feed used on pig farms, where it was detected in up to 50% of samples of various animal feeds [30], so that the possibility of carry-over effect during pig meat products’ use should be further investigated. There is also a study on STC occurrence in spices, such as pepper (up to 125 µg/kg) and red pepper (up to 23 μg/kg), likely to be used in dry-fermented meat products’ production [14].

In the recent study of mycotoxin contamination of the traditional sausage *Kulen* performed by Lešić et al. [27], OTA concentrations in 2 out of 16 samples exceeded the recommended value of 1 µg/kg in pork meat and products adopted by the Italian Ministry of Health (3.91 and 6.95 µg/kg) [37]. In that research, in most of the OTA-contaminated samples, no OTA-producing mould was identified. It was therefore concluded that except for direct mould production, OTA presence might be attributed to the use of contaminated spices, for instance black and red peppers. In this study, 7 out of 24 (29%) OTA concentrations determined in the investigated samples exceeded the above-mentioned recommended OTA value. Of note, OTA was mostly present in dry-fermented sausages, not dry-cured meat products (Appendix A). In other studies that investigated OTA incidence in meat products in the last 10 years, the maximum detected OTA concentrations were <LOD—12.48 µg/kg [13,15,31,38,39,40]. The highest concentration determined in this study was 4.81 µg/kg in prosciutto (Appendix A). As a rule, in every study in which OTA presence was uncovered, at least one product in which its concentration exceeded the limit was found.

Studies on CPA presence in foods and feeds, which are few in number, have shown that the latter mycotoxin can be found in peanuts, corn, rice, dry figs, tomato puree, sunflower, wheat, and fermented products of animal origin, such as cheese and dry-cured meat products [24]. Although *P. commune*, a significant CPA producer, is one of the predominant mould species isolated from dry-fermented meat products, CPA concentrations in products of this type are generally unexplored [26,41,42]. To the best of our knowledge, data published so far include 1 study of Spanish dry-cured hams, in which CPA concentrations ranged from 36 to 540 µg/kg [28], and 2 studies of Croatian traditional sausages [26,27], in which CPA was found in 15% and 31% samples, in the concentration range of 2.50–59.80 µg/kg. In comparison with previous research, the results of this study, which included dry-fermented sausages and dry-cured meat products, showed higher CPA concentrations than in the previous two Croatian sausage studies, and slightly lower concentrations than those found in Spanish dry-cured hams. In this study, CPA was similarly represented in dry-fermented sausages and dry-cured meats, with the highest concentrations determined in one sausage (335.5 µg/kg) and one prosciutto sample (108.5 µg/kg) (Appendix A).

### 2.3. Mycotoxins in Dry-Cured Meat Products in Relation to the Region of the Product’s Origin and the Weather

Environmental conditions affect mycotoxin production to a greater extent than mould growth. Among other factors, this production depends on weather in a particular geographical area that varies on an annual basis [33].

The incidence of mycotoxins across the Croatian production regions (central, eastern, northern, western and southern) and sampling years is shown in Table 2. Although no statistically significant differences (*p* > 0.05) in the concentrations of individual mycotoxins were found between either the sampling years (CPA *p* = 0.184; OTA *p* = 0.976) or the production regions (CPA *p* = 0.551; OTA *p* = 0.485; STC *p* = 0.948), there was an evident difference in mycotoxin occurrence.

Maps representing the average temperatures and precipitation in Croatia during the ripening of the meat products sampled in 2020 and 2021, and the mean OTA, STC, and CPA concentrations determined in the analysed meat products from five Croatian regions (and their respective districts) are shown in Figure 2a–f, Figure 3a–f and Figure 4a–f.

The eastern (20) and western (18) regions delivered the most contaminated samples, while the southern (5) region delivered the fewest. As for each mycotoxin, the largest percentage of samples in which CPA was detected was from the eastern region (41%) followed by the western (26%), while the smallest number of such samples came from the southern region (6%). The highest OTA occurrence was seen in the central (29%), western (25%), and eastern (25%) regions, with the lowest occurrence seen in the southern region (4%). The incidence of STC was about the same in all regions (4–8%) but the eastern, in which the STC presence was not detected (Figure 3). The highest (although not statistically significant) STC concentrations were determined in the northern region (1.25 ± 1.81 µg/kg). The highest average OTA concentration was found in samples originating from the western region (1.45 ± 1.67 µg/kg) and in one sample from the southern region (2.66 µg/kg). The highest average CPA concentration was determined in samples coming from the eastern region (36.25 ± 86.71 µg/kg) and one sample retrieved from the southern region (108.50 µg/kg) (Table 2).

The eastern and the western regions have similar moderate climates, while the southern region is often compared with tropical and subtropical regions, where more *Aspergillus* than *Penicillium* species are commonly identified [6,43]. This is attributed precisely to the higher temperatures and dryness preferred by the *Aspergillus* species, whose spores are more resistant to dry air and high temperatures, while lower ambient temperatures are more suitable for *Penicillium* species [44]. Therefore, OTA and CPA detected in these moderate climate regions are probably mostly produced by *Penicillium* rather than *Aspergillus* species.

The western region, where the highest average OTA concentrations were detected (except for one vastly contaminated sample from the southern region), is characterized by moderate precipitation and an average temperature of 15 °C (Figure 2), optimal for OTA production by *P. nordicum*. *P. nordicum* has been shown to produce OTA in the temperature range of 15–30 °C [13]. The eastern region, where the highest average CPA concentration was found (again, except for one vastly contaminated sample from the southern region), is characterized by low precipitation and an average temperature of 10–12 °C (Figure 4). Sosa et al. [41] showed that *P. commune* can grow at each tested temperature (12–30 °C), but the optimal growth-enhancing conditions were temperatures of 20–25 °C and a_w_ of 0.96, while the most CPA was produced at 30 °C and a_w_ of 0.96. The above supports the CPA values determined in this study during warm weather [45]. One of the highest CPA concentrations determined in this study was the one in a sample from the southern region, where the average daily temperature in the warmest months often exceeds 35 °C. Sosa et al. [41] concluded that the temperature represents the most important factor influencing CPA production, although the interactions of various factors such as temperature and a_w_ are also important.

As for the season in which the STC-positive products were ripened, it should be pointed out that unlike in other sampling regions, products coming from the eastern region had ripened only during winter, that is, at lower temperatures; in other regions, the products had ripened in spring/summer, i.e., at higher temperatures, which is more favourable for the *Aspergillus* species, the main STC producers, in particular for the *Versicolores* section [46]. The usual low temperatures in eastern Croatia during winter months provide for good natural meat processing environments. For example, the average temperature in Slavonia from November to March ranges from 0.2 to 6.9 °C [47].

In general, more OTA- and STC-contaminated samples were identified in the first sampling year (2020) than in the second (2021); conversely, more CPA-contaminated samples were determined in the second sampling year. According to the Official National Weather Reports, within the 2019–2021 timeframe, i.e., within the meat products’ production period, the environmental temperatures all over Croatia could be described as high/very high, while data on the amount of precipitation show that the study period could be described as rainy/characterised with a normal precipitation amount [45]. It is therefore difficult to draw conclusions on the impact of weather on mycotoxin production by toxigenic mould species. That is, these production environments can be very hot but also humid should the weather be rainy. High temperature and humidity are generally considered to increase the risk of mould growth and mycotoxin production, with warmer and drier weather thereby being better suited for *Aspergillus* and colder and rainier weather better for *Penicillium* species growth.

Evidence indicates that European countries with moderate climates are more exposed to moulds and mycotoxins due to climate change [30,48]. The climates in these countries are likely to become warmer, reaching the temperature of 33 °C, a temperature very close to optimal for *A. versicolores* (30 °C) growth and STC production (optimum temperature range: 23–29 °C) [30]. Climate changes consequent to the global warming trend have also been witnessed in Croatia in recent decades, with the weather significantly deviating from its long-term patterns [49]. Examples of modified weather patterns affecting mycotoxins were shown during the summer seasons of 2003 and 2004 and then in 2012 in the Mediterranean region, where drought and elevated temperatures resulted in a significant contamination of maize with *A. flavus* and aflatoxins, and aflatoxin M_1_ entered the dairy chain through the feed chain [33,50]. The 2009–2011 analysis of maize samples originating from Serbia did not detect aflatoxins, but due to the prolonged hot and dry weather in 2012, the subsequent analysis resulted in 69% aflatoxin-contaminated samples [51].

In this study, mycotoxins AFB_1_ and CIT were not detected in any of the analysed samples, although their producers, *A. flavus* and *P. citrinum*, are often isolated from Croatian dry-cured meat products [6,43]. This confirms that the growth of toxigenic moulds does not necessarily imply the presence of mycotoxins in food since their production depends on a number of environmental factors [32]. Research shows that environmental factors (such as the combination of temperature × a_w_ × CO_2_) directly affect the expression of AFB_1_ biosynthetic genes but have no significant impact on *A. flavus* growth [28]. In other studies that investigated the AFB_1_ contamination of meat products in the last 10 years, this mycotoxin was rarely detected, always in concentrations < LOD—1.91 µg/kg [13,31,36].

Bailly et al. [42] reported relatively high amounts of CIT produced in dry-fermented meat products by the *P. citrinum* toxigenic strain after 16 days of incubation at 20 °C. According to the current knowledge, there is only one study of CIT occurrence in Croatian meat products, which made use of the ELISA assay and detected CIT in concentrations of around LOD in 4.4% of samples [31]. There are also studies on CIT occurrence in feed (in which it was found in concentrations of up to 998 µ/kg), demonstrating a possible carry-over effect [23], as well as on its presence in spices such as black pepper (in which it was found in concentrations of up to 50 µg/kg) [25] that are likely to be used in dry-fermented meat products’ production. The study of feed samples from pig farms that reported nephropathy in livestock found CIT in 96% of samples, with the mean concentration of 120.5 ± 43.3 μg/kg [52].

Weather reports given in this study cover the ripening period of all products in which mycotoxins were detected. Generally, it is a long period, 1.5 years, but some products mature for only a few months in which the weather might differ from the annual average, making the reliable assessment of the influence of weather on mycotoxin occurrence in these products rather difficult. On top of the above, unlike plants or cereals, which are directly exposed to weather and subsequent mould growth, dry-cured meat products mature in chambers in which some of the environmental factors can be regulated, although this is regrettably not customary for home-based productions. TMPs can be contaminated with mycotoxins not only via direct exposure but also through contaminated raw materials or due to the carry-over effect from contaminated feed [18].

## 3. Conclusions

The investigated mycotoxins were detected in 27% of 250 analysed meat product samples, with the highest occurrence and concentrations of CPA (13% of samples in concentrations of up to 335.5 µg/kg), followed by OTA (10% of samples) and STC (4% of samples). No contamination with AFB_1_ and CIT was observed. In 29% of OTA-positives, the determined concentrations exceeded the Italian guidance value for OTA in pork meat. The study revealed a rather high percentage of positive dry-fermented sausages in comparison with dry-cured meat products, indicating a possible contamination of spices added to the sausages. Therefore, the control of ingredients used in the production of these meat products should not be neglected. Mycotoxin contamination is related to regional weather during production, which can generally be characterized as warm and rainy and therefore increases the risk of mould growth and mycotoxin production. Although no statistically significant differences in the concentrations of individual mycotoxins were found across the production regions, differences in mycotoxin occurrence was found. The eastern and the western regions, having moderate climates, delivered the most contaminated samples, while the southern region, often compared with subtropical regions, delivered the fewest such samples; therefore, the determined mycotoxins were probably mainly produced by the *Penicillium* rather than the *Aspergillus* species. Due to the interactions of various factors that may affect mycotoxin biosynthesis during production, the detected concentrations cannot be solely related to the weather. The obtained results indicate the necessity of further research into all potential sources and mechanisms of contamination during all stages of TMP production and storage that are potentially responsible for the occurrence of mycotoxins in these types of products.

## 4. Materials and Methods

### 4.1. Dry-Cured Meat Products

Croatian traditional dry-cured meat products (*n* = 250), composed of dry-fermented sausages (domestic sausages, *Kulen*, and *Kulenova Seka*) and dry-cured meats (prosciutto, dry-cured hams, dry rack, *Pečenica*, *Ombolo*, bacon and pancetta) were sampled on the producing family farms in the amount of 1.5–2.0 kg in full line with the regulation [53] defining food sampling procedures for the analytical determination of mycotoxins. The samples originated from family farms seated in five Croatian regions (the eastern region, embracing Vukovar-Srijem, Osijek-Baranja, Virovitica-Podravina, and Požega-Slavonia districts, *n* = 52 samples; the northern region, embracing Koprivnica-Križevci, Varaždin, and Virovitica-Podravina districts, *n* = 62 samples; the central region, embracing Krapina-Zagorje and Zagreb districts, *n* = 24; the western region, embracing Istria and Primorje-Gorski Kotar districts, *n* = 55; and the southern region, embracing Split-Dalmatia, Dubrovnik-Neretva, and Šibenik-Knin districts, *n* = 57).

Home-based production is characterised by poorly controlled and variable production conditions and seasonality and makes no use of bacteria or mould starter cultures. Dry-fermented sausages are produced from minced pork meat and fat, supplemented by salt and other spices and stuffed into casings, while dry-cured meats are produced from bony or boneless pork meat and subcutaneous and cutaneous tissue and spiked with salt. Dry-cured meat products are then dried and left to ripen either smoked or unsmoked. Drying and ripening take place mostly during winter, but some products, such as hams and prosciuttos, ripe throughout the whole year in ripening chambers conditioned to maintain a temperature below 16 °C and relative humidity of 65–85%. During ripening, the surface moulds are continuously rinsed and brushed off so as to prevent excessive mould on the final product. 

The sampling was carried out over a two-year period (2020–2021). In each sampling year, on three separate occasions (in spring, summer, and autumn depending on the maturation of the targeted products), roughly the same quantities of the same types of samples were taken from the same localities whenever possible. During the sampling, data on the durations of product maturation were recorded so as to relate them to the weather. Ripening duration depends on the product type (for example, from 2 months for some dry-fermented sausages to 17 months for some prosciuttos). Ripening duration depends on the product type (for example, from 2 months for some dry-fermented sausages to 17 months for some prosciuttos). Samples were homogenized using a Grindomix GM 200 homogenizer (Retsch, Haan, Germany).

### 4.2. Chemicals

CPA (Art. No. C 1530) and AFB_1_ (Art. No. A6636) analytical standards were purchased from Sigma-Aldrich (St. Louis, MO, USA). OTA (Art. No. AC227400050), STC (Art. No. 10048-13-2), and CIT (Art. No. 518-75-2) standards were obtained from LGC Standards (Wesel, Germany). CPA, STC, and AFB_1_ stock solutions were prepared by dissolving 1 mg of the standard in 10 mL of acetonitrile (100 μg/mL). OTA and CIT standards were obtained by dissolution in acetonitrile, in concentrations of 10 μg/mL and 100 μg/mL, respectively. Ultrapure water was obtained from a Direct-Q 3 UV device (Merck, Darmstadt, Germany). Liquid chemicals were obtained from Honeywell (Charlotte, NC, USA) and solid from Sigma-Aldrich (St. Louis, MO, USA).

### 4.3. Preparation of the Samples for LC-MS/MS Analysis

The preparation of samples intended for AFB_1_, OTA, STC, and CIT analysis involved the use of highly specific immunoaffinity columns (R-Biopharm Rhône Ltd., Glasgow, Scotland) and was carried out according to the manufacturer’s instructions. AFB_1_ and OTA analysis employed AFLAOCHRAPREP^®^ columns, while CIT analysis made use of Easi-extract CITRININ^®^ and STC Easi-extract STERIGMATOCYSTIN^®^ columns. The procedure is described in detail by Lešić et al. [22].

The preparation of samples for the CPA analysis was described in detail by Vulić et al. [26] and involved the use of rOQ QuEChERS Extraction Packets (Phenomenex, Torrance, CA, USA) and Captiva EMR-Lipid SPE columns (Agilent Technologies, Santa Clara, CA, USA); the latter were used for fatty samples such as sausages and bacon.

### 4.4. Mycotoxin Analysis by the LC-MS/MS

LC–MS/MS analysis was performed using a high-performance liquid chromatograph (1260 Infinity, Agilent Technologies, Santa Clara, CA, USA) coupled with a triple quadrupole mass spectrometer equipped with an electrospray ionization (ESI) source (6410 QQQ, Agilent Technologies, Santa Clara, CA, USA). The chromatographic separation of mycotoxins was performed on a 150 mm × 4.6 mm, 5 µm particle size Gemini analytical column (Phenomenex, Torrance, CA, USA) coupled with a SecurityGuard^TM^ Cartridges Gemini^®^ C18, 4 mm × 3.0 mm ID pre-column (Phenomenex, Torrance, CA, USA).

The chromatographic and instrumental mass spectrometry settings for CIT and STC analysis as well as for AFB_1_ and OTA analysis were the same, as previously described by Kudumija et al. [36]. Chromatographic and instrumental mass spectrometry employed with the CPA analysis was described by Vulić et al. [26]. Precursor and product ions with other parameters obtained during optimization are shown in Table 3.

### 4.5. Method Validation

The LC–MS/MS method was validated according to the guidance document [54]. Each mycotoxin was validated separately. Ten different blank sausage samples (five different types in two replicates) were spiked with 0.1 µg/kg of STC, 0.1 µg/kg of AFB_1_, 0.3 µg/kg of OTA, 2 µg/kg of CIT, and 3 µg/kg of CPA for LOD/LOQ determination, prepared and analysed. For each batch, a 5-point calibration curve was plotted. The concentration ranges were as follows: STC 0.1, 0.5, 2.5, 5, and 10 ng/mL; OTA, 0.2, 0.4, 1.0, 2.5, and 5 ng/mL; AFB_1_, 0.05, 0.1, 0.25, 0.5, and 1 ng/mL; CIT 1.0, 2.0, 4.0, 8.0, and 10 ng/mL; and CPA, 0.5, 2.5, 10, 15, and 25 ng/mL. LOD and LOQ were determined based on the calibration curves’ slopes and the signal abundances of the spiked samples. Linearity was tested within the concentration ranges quoted above, while the recovery was determined by analysing 10 different blank sausage samples (5 types s in two replicates) spiked with 0.1 µg/kg of STC, 0.1 µg/kg of AFB_1_, 0.3 µg/kg of OTA, 2 µg/kg of CIT, and 3 µg/kg of CPA. Since the solvent calibration was used, the matrix effect was evaluated by comparing the peak areas of each mycotoxin in the standard solution at 0.25 ng/mL for AFB_1_, 0.50 ng/mL for STC, 0.40 ng/mL for OTA, 2 ng/mL for CIT, and 1 ng/mL for CPA and the blank matrix spiked after sample preparation at the same level.

### 4.6. Meteorological Data

Data on the weather (environmental temperatures and precipitation amounts) in Croatia in the sampling years 2020 and 2021 (during the meat products’ ripening) were obtained from the Croatian Meteorological and Hydrological Services. The products sampled in the first sampling year (2020) were produced and ripened in 2019, the first half of 2020, while the products sampled in the second sampling year (2021) were produced and ripened in 2020, the first half of 2021. Based on the average monthly air temperatures (°C) and precipitation amounts (mm) obtained from weather stations in the districts where these traditional meat products were produced, maps were designed using Google Fusion Tables (Google, Mountain View, CA, USA).

### 4.7. Data Analysis

Statistical analyses were performed using SPSS Statistics 22.0 (IBM, Armonk, NY, USA). The results were tested for distribution normality using the Shapiro–Wilks test. In order to determine the statistical significance of the differences in the detected mycotoxin concentrations across the sampling years and production regions, the Mann–Whitney U test and Kruskal–Wallis test were used. Decisions on statistical relevance were made at the significance level of *p* < 0.05.

## Figures and Tables

**Figure 1 toxins-14-00476-f001:**
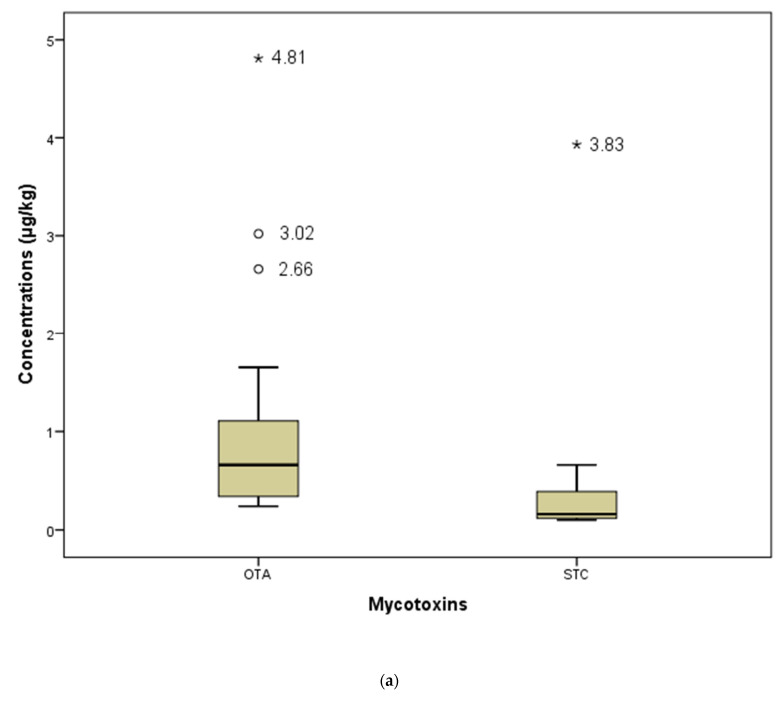
Box plot of mycotoxin concentrations (>LOD) in dry-cured meat products: (**a**) OTA and STC (**b**) CPA.

**Figure 2 toxins-14-00476-f002:**
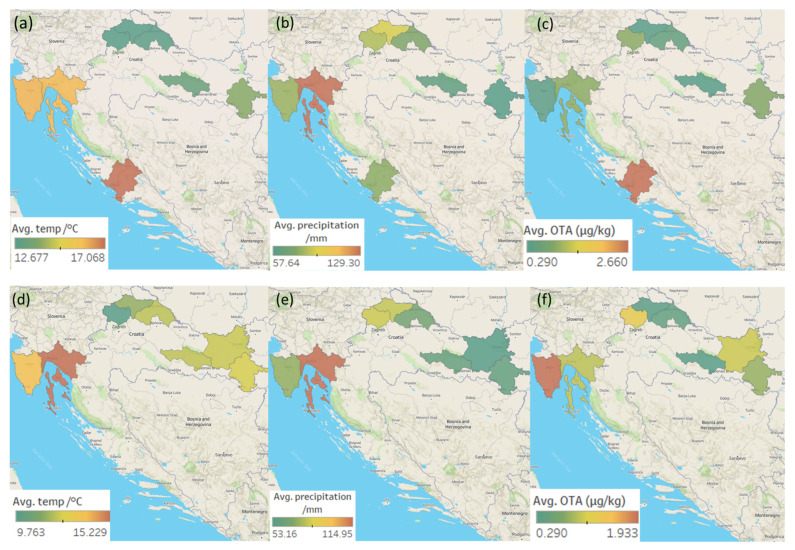
Maps representing the regional weather and average OTA concentrations in the five Croatian production regions of traditional meat products in the 1st sampling year (ripening: 2019—the first half of 2020): (**a**) average temperatures, (**b**) average precipitation; (**c**) average OTA concentrations in positive samples. Regional weather and average OTA concentrations in the 2nd sampling year (ripening: 2020—the first half of 2021): (**d**) average temperatures; (**e**) average precipitation; (**f**) average OTA concentrations in positive samples.

**Figure 3 toxins-14-00476-f003:**
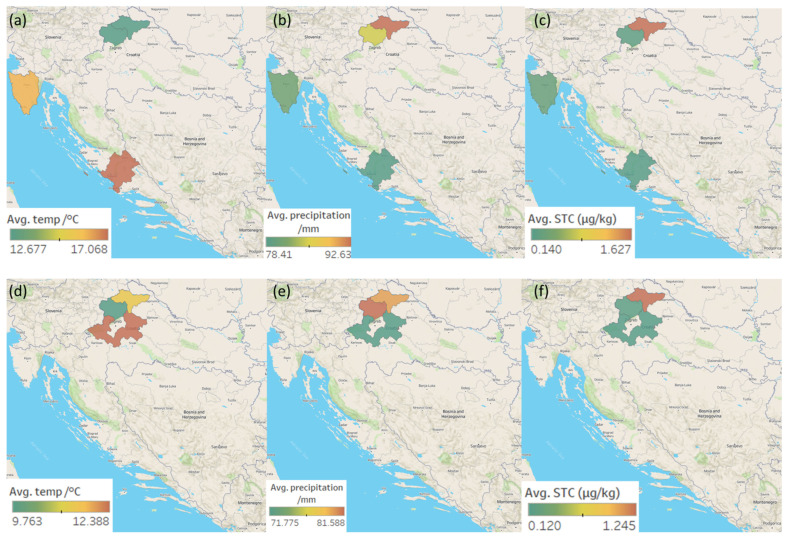
Maps representing regional weather and average STC concentrations in five Croatian production regions of traditional meat products in the 1st sampling year (ripening: 2019—the first half of 2020): (**a**) average temperatures; (**b**) average precipitation; (**c**) average STC concentrations in positive samples. Regional weather and average OTA concentrations in the 2nd sampling year (ripening: 2020—the first half of 2021): (**d**) average temperatures; (**e**) average precipitation; (**f**) average STC concentrations in positive samples.

**Figure 4 toxins-14-00476-f004:**
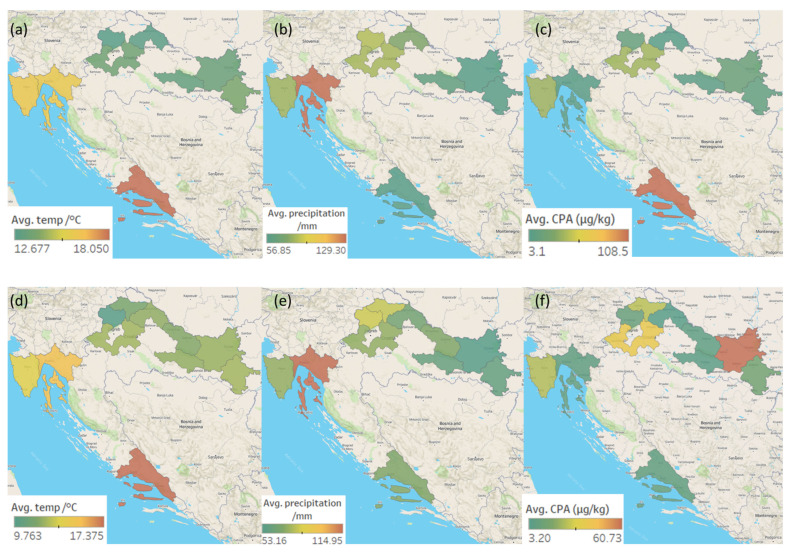
Maps representing regional weather and average CPA concentrations in the five Croatian production regions of traditional meat products in the: 1st sampling year (ripening: 2019—the first half of 2020): (**a**) average temperatures; (**b**) average precipitation; (**c**) average CPA concentrations in positive samples. Regional weather and average OTA concentrations in the 2nd sampling year (ripening: 2020—the first half of 2021): (**d**) average temperatures; (**e**) average precipitation; (**f**) average CPA concentrations in positive samples.

**Table 1 toxins-14-00476-t001:** Validation of the use of LC–MS/MS to analyse the mycotoxins in dry-cured meat products.

Mycotoxins	Limit of Detection (μg/kg)	Limit of Quantification (μg/kg)	Recovery (%)	Matrix Effect (%)
OTA	0.18	0.59	119.4	1.8
AFB_1_	0.03	0.11	91.4	6.4
CPA	2.45	8.07	97.52	1.9
CIT	0.60	1.98	100.9	0.7
STC	0.02	0.06	114.4	7.7

*n* = 10; OTA—ochratoxin A; AFB_1_—aflatoxin B_1_; CPA—cyclopiazonic acid; CIT—citrinin; STC—sterigmatocystin.

**Table 2 toxins-14-00476-t002:** Descriptive statistics pertaining to the mycotoxin occurrence across the production regions and sampling years.

Mycotoxin	Sampling Year	Croatian Regions														
		Southern (*n* = 57)			Western (*n* = 55)			Eastern (*n* = 52)			Northern (*n* = 62)			Central (*n* = 24)		
		*n*%/*n*	Mean ± SD (Min–Max) µg/kg	Med	*n*%/*n*	Mean ± SD (Min–Max) µg/kg	Med	*n*%/*n*	Mean ± SD (Min–Max) µg/kg	Med	*n*%/*n*	Mean ± SD (Min–Max) µg/kg	Med	*n*%/*n*	Mean ± SD (Min–Max) µg/kg	Med
AFB_1_	2020–2021	<LOD	<LOD	<LOD	<LOD	<LOD	<LOD	<LOD	<LOD	<LOD	<LOD	<LOD	<LOD	<LOD	<LOD	<LOD
OTA	2020	4/1	2.66 ± na	na	15/4	0.82 ± 0.36 (0.75–1.17)	0.88	14/4	0.76 ± 0.66 (0.24–1.65)	0.57	10/3	0.43 ± 0.24	0.29	27/3	0.97 ± 0.56 (0.61–1.61)	0.68
	2021	0/0	<LOD	na	7/2	2.73 ± 2.95 (0.64–4.81)	2.73	8/2	0.78 ± 0.37 (0.51–1.04)	0.78	3/1	0.36 ± na	na	31/4	1.02 ± 1.34 (0.27–3.02)	0.40
	2020–2021	2/1	2.66 ± na	na	11/6	1.45 ± 1.67	0.88	12/6	0.76 ± 0.54	0.68	7/4	0.41 ± 0.20	0.33	29/7	1.00 ± 1.00	0.61
CIT	2020–2021	<LOD	<LOD	<LOD	<LOD	<LOD	<LOD	<LOD	<LOD	<LOD	<LOD	<LOD	<LOD	<LOD	<LOD	<LOD
STC	2020	8/2	0.14 ± 0.03 (0.12–0.16)	0.14	12/3	0.29 ± 0.18 (0.13–0.49)	0.24	<LOD	<LOD	<LOD	10/3	1.63 ± 2.00 (0.29–3.93)	0.66	0/0	<LOD	<LOD
	2021	0/0	<LOD	<LOD	0/0	<LOD	<LOD	<LOD	<LOD	<LOD	3/1	0.10 ± na	na	15/2	0.12 ± 0.01 (0.11–0.12)	0.12
	2020–2021	4/2	0.14 ± 0.03	0.14	6/3	0.29 ± 0.18	0.24	<LOD	<LOD	<LOD	7/4	1.25 ± 1.81	0.10	8/2	0.12 ± 0.01	0.12
CPA	2020	4/1	108.50 ± na	na	12/3	23.57 ± 37.06 (1.25–66.35)	3.10	21/6	11.76 ± 12,59 (3.55–36.50)	6.18	7/2	21.23 ± 25.49 (3.20–39.25)	21.23	18/2	6.51 ± 8.90 (0.21–12.80)	6.51
	2021	3/1	6.75 ± na	na	21/6	19.98 ± 15.06 (6.25–44.70)	17.28	33/8	54.63 ± 113.80 (4.20–335.50)	18.60	13/4	21.59 ± 18.39 (6.10–44.25)	18.00	8/1	5.55 ± na	na
	2020–2021	4/2	57.63 ± 71.95	57.63	16/9	21.17 ± 22.10	9.90	27/14	36.25 ± 86.71	9.95	10/6	21.47 ± 18.25	18.00	13/3	6.19 ± 6.32	5.55

LOD—limit of detection; na—not applicable; *n*%/*n*—percentage/number of samples above LOD; Med—median. OTA—ochratoxin A; AFB1—aflatoxin B1; CIT—citrinin; STC—sterigmatocystin; CPA—cyclopiazonic acid.

**Table 3 toxins-14-00476-t003:** Instrumental LC–MS/MS settings.

Analyte	Precursor Ion	Fragmenter Voltage (V)	Product Ions	Collision Energy (eV)
CIT	251.1	110	233.1	15
			205.0	25
STC	325.1	130	310.0	25
			281.0	40
OTA	404.0	130	357.9 239.0	25 10
AFB_1_	313.1	170	285.1 269.1	23 30
CPA	337.2	110	196.3 182.1	25 20

CIT—citrinin; STC—sterigmatocystin; OTA—ochratoxin A; AFB_1_—aflatoxin B_1_; CPA—cyclopiazonic acid.

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
