# Peer review of "The Occurrence of Five Unregulated Mycotoxins Most Important for Traditional Dry-Cured Meat Products"

_toxins, 2022, doi:10.3390/toxins14070476_

Round 1
Reviewer 1 Report
General comments
This manuscript was described the relationship of mycotoxin occurrence in traditional dry-cured meat products and the origin a local weather.
It is interesting attempt to analyze climate change and mycotoxin contamination by focusing on the traditional fermented foods of the region.
However, as the author also mentioned, the origin of the mycotoxin contaminating the TMP cannot be attributed to the molding that takes place in the area. Most likely derived from feed and spices. It is not scientific to compare the data with climate change data without a clear origin of mycotoxin contamination.
If the originality of this paper is to be enhanced, it would be necessary to examine the molds that affect the mold attachment of TMPs and their ability to produce fungal toxins.
1.Introduction
Please abbreviate by stating the FULL name of the TMP the first time it appears.
2.Figs and Tables
Tables 1 and 2 should be listed together and Figure S1 should be placed in the text.
3. Results and Discussion
Mycotoxin concentration in TMP are sometime extreme, but the basis for this is not clear.
Author Response
COMMENT: This manuscript was described the relationship of mycotoxin occurrence in traditional dry-cured meat products and the origin a local weather. It is interesting attempt to analyze climate change and mycotoxin contamination by focusing on the traditional fermented foods of the region. However, as the author also mentioned, the origin of the mycotoxin contaminating the TMP cannot be attributed to the molding that takes place in the area. Most likely derived from feed and spices. It is not scientific to compare the data with climate change data without a clear origin of mycotoxin contamination. If the originality of this paper is to be enhanced, it would be necessary to examine the molds that affect the mold attachment of TMPs and their ability to produce fungal toxins.
ANSWER: We agree with the Reviewer’s comment that the identification of moulds that overgrow TMP surfaces and the assessment of their ability to produce mycotoxins is of great importance. Identification of moulds and their toxicogenic species are in the focus of our research conducted within the frame of a four-year science research project. Among other, the Project includes identification of moulds growing on the surfaces of traditional meat products and investigates into the link between mycotoxin presence and toxicogenic species, determining thereby all genes responsible for mycotoxin production. However, we are of the opinion that, on top of the results already presented in this contribution, the display of all mould-related research results (including toxicogenic mould species and genes responsible for mould production in each and every type of meat products investigated insofar across various production regions) would make the content of this single contribution far too abundant. We have every intention to publish that research segment in our forthcoming publication, which will look into the mould identification in light of the results presented herein. In line with the Reviewer’s next comment, the Conclusion section has been supplemented with the sentence stating that “the obtained results indicate the necessity of further research into all potential sources and mechanisms of contamination during all stages of TMP production and storage, potentially responsible for the occurrence of mycotoxins in these types of products”.
COMMENT: Introduction. Please abbreviate by stating the FULL name of the TMP the first time it appears.
ANSWER: We are grateful to the Reviewer for this remark. In the Introduction section abbreviation TMP is now explained in full (“traditional meat products”), while the abbreviation “TMP” is given in brackets.
COMMENT: Figs and Tables. Tables 1 and 2 should be listed together and Figure S1 should be placed in the text.
ANSWER: According to the Reviewer’s recommendation, Tables 1 and 2 are now listed together. Instead of being a supplementary material, the former Figure S1 is now incorporated into the body text as Figure 1.
COMMENT: Results and Discussion. Mycotoxin concentration in TMP are sometime extreme, but the basis for this is not clear.
ANSWER: We are most obliged to the Reviewer for his/her most useful comment. Extreme mycotoxin concentrations were mostly established for cyclopiazonic acid (CPA), which has already been proven to be highly stable when in meat products, in which it tends to accumulate in very high concentrations (Bailly, J.D., Guerre, P. 2009. Mycotoxins in meat and processed meat products. In: F. Toldrá (ed.), Food microbiology and food safety - Safety of meat and processed meat, New York: Springer ). This research proves this hypothesis right and applicable to the Croatian TMPs, as well. Circumstances that might have been responsible for such an extreme contamination are to be established in our future research. Namely, for the purpose of this study, meat product samples had been retrieved from family farms that produce them in order to sell, so that the primary study goal was to investigate into the presence of mycotoxins most relevant for this type of foodstuffs, establish their concentrations, and link their presence with the local weather typical of the production region. Our research envisaged within the Project frame is ongoing, and should, beyond doubt, include the identification of TMP contamination routes and mechanisms, as well as the identification of all circumstances capable of causing such a contamination. In light of both this and the very first comment given by the Reviewer, the Conclusion section has been supplemented with the sentence stating that “the obtained results indicate the necessity of further research into all potential sources and mechanisms of contamination during all stages of TMP production and storage, potentially responsible for the occurrence of mycotoxins in these types of products”.
The authors are indebted to our esteemed Reviewer for helpful suggestions and comments.
Reviewer 2 Report
The topic of this manuscript is very important from the point of view of food safety, its production as well as food analysis. Thematically, this manuscript fits within the scope of Toxins journal, therefore I believe it must be approved for publication in this journal.
The presentation and discussion of the results do not raise my reservations, I have no comments on these parts of the manuscript. In my opinion, the text should be slightly improved, my suggestions are as follows:
· The section “4. Materials and Methods ”- please describe in detail the dry-cured meat product samples - were these products maturing spontaneously or with the use of starter cultures? What were the maturation conditions (parameters)? What were the sanitary conditions of the production and maturation process?
· The section „Abstract” – in my opinion, this fragment of the manuscript should cover all aspects of the obtained results, including the correlation of mycotoxins with the analyzed atmospheric conditions. I suggest modifying the Abstract a bit.
Author Response
COMMENT: The topic of this manuscript is very important from the point of view of food safety, its production as well as food analysis. Thematically, this manuscript fits within the scope of Toxins journal, therefore I believe it must be approved for publication in this journal.
ANSWER: We thank the Reviewer for this comment and the appreciation of the subject-matter relevance.
COMMENT: The presentation and discussion of the results do not raise my reservations, I have no comments on these parts of the manuscript. In my opinion, the text should be slightly improved, my suggestions are as follows: The section “4. Materials and Methods”- please describe in detail the dry-cured meat product samples - were these products maturing spontaneously or with the use of starter cultures? What were the maturation conditions (parameters)? What were the sanitary conditions of the production and maturation process?
ANSWER: In the revised Materials and Methods section, the production of these traditional meat products has now been described in much more detail. The supplemented text now informs the readership that these products are produced without the use of starter cultures, under semi-controlled conditions where maturation time depends on the product type (for example, from 2 months for some dry-fermented sausages to 17 months for some prosciuttos). The contribution now contains the description of all production conditions known to the authors, but regrettably not the sanitary conditions in the production environments which we are unaware of, since the study dealt with final products (i.e., those already produced and intended for either consumption or market release).
COMMENT: The section „Abstract” – in my opinion, this fragment of the manuscript should cover all aspects of the obtained results, including the correlation of mycotoxins with the analyzed atmospheric conditions. I suggest modifying the Abstract a bit.
ANSWER: We are grateful to the Reviewer for this very helpful comment. The Abstract has been revised and supplemented with the sentence about mycotoxin occurrence in relation to the weather, which hopefully added to the Abstract comprehensiveness.
The authors are indebted to our esteemed Reviewer for helpful suggestions and comments.
Reviewer 3 Report
The manuscript entitled “The Occurrence of Mycotoxins in Traditional Dry-Cured Meat Products relative to the Region of Origin and Local Weather” evaluated the load of mycotoxin in the traditional homemade dry-fermented sausages and dry-cured meat products. I found the study very interesting. However, I highly recommend the manuscript to go through a major revision and address the following questions:
1. Out of so many mycotoxins why authors have decided to evaluate only Aflatoxin B1, ochratoxin A, sterigmatocystin, citrinin, and cyclopiazonic acid?
2. I could see that authors did not detect AFB1 and CIT in any of the TMP samples. There are several types of Aflatoxins, did authors also not find any of them?
3. I think authors should mention the time frame for the sample collection.
4. What are the current legislation limits of these mycotoxins in food products?
5. I recommend presenting the data in a form of table indicating the type of mycotoxin detected and its amount in different types of samples. For example, how many samples of domestic sausages, cured hams, bacon, dry rack were collected and what is the level of mycotoxin detected in it?
6. Is the contamination occurring during the processing of meat preservation and packaging or is it present in the animal body, which may also come from the feed they were given?
7. I strongly recommend improving the introduction and discussion section of the manuscript. Write how these mycotoxins are incorporated in the food chain and being toxic to the living organisms present in the environment. The below mentioned papers are suitable for citation:
Deepshikha et al., 2022. Front Nutr. 9:851787.
Dey et al. 2021. Environ Pollut. 268.115713.
Dey et al. 2022. Crit Rev Food Sci Nutr. 1-22.
Author Response
COMMENT: The manuscript entitled “The Occurrence of Mycotoxins in Traditional Dry-Cured Meat Products relative to the Region of Origin and Local Weather” evaluated the load of mycotoxin in the traditional homemade dry-fermented sausages and dry-cured meat products. I found the study very interesting. However, I highly recommend the manuscript to go through a major revision and address the following questions:
ANSWER: We thank the Reviewer for the recognition of the importance of the research area covered by this contribution.
COMMENT: Out of so many mycotoxins why authors have decided to evaluate only Aflatoxin B1, ochratoxin A, sterigmatocystin, citrinin, and cyclopiazonic acid?
ANSWER: Earlier studies showed that aflatoxin B1 (AFB1) and ochratoxin A (OTA) can be found in traditional meat products (Pleadin et al., 2015. Food Control 52, 71-77), and represent the mycotoxins of the greatest public concern due to their toxicity. Insofar, other mycotoxins potentially present in meat products have been only poorly investigated. Sterigmatocystin (STC) is the AFB1 precursor and was therefore also included into this study. Furthermore, the co-occurrence of AFB1 and STC, or AFB1 and CPA, in some types of foods was already reported in the literature (Ostry et al., 2018. World Mycotoxin J. 11, 135–148). The same goes for the co-occurrence of OTA and citrinin (CIT) (EFSA J. 2012, 10, 2605). Given that mould species predominantly found on TMP surfaces are P. citrinum and P. commune (Zadravec et al., 2020. Int. J. Food Microbiol. 317, 108459; Lešić et al., 2021. Processes 9,1-15) known to be CIT- and cyclopiazonic acid (CPA)-producers, and given that their occurrence in meat products is still underexplored despite of the fact that some authors claim that meat products are precisely the type of foodstuffs in which the above mycotoxins are expected to be found, our research dealt with these five mycotoxins believed to be of the greatest interest when it comes to meat products. To the best to our knowledge, and according to the literature, there is no indication that other mycotoxins potentially produced by moulds of the Aspergillus and the Penicillium genera, are associated with dry-cured meat products’ contamination.
COMMENT: I could see that authors did not detect AFB1 and CIT in any of the TMP samples. There are several types of Aflatoxins, did authors also not find any of them?
ANSWER: We are grateful to the Reviewer for this particular question. On top of AFB1, LC-MS/MS analytics applied within this study frame also aimed to detect/quantify AFB2, AFG1 and AFG2, so that the samples were also analysed for each of those, but none of them was detected in any of the TMP samples. The pertaining results have been omitted from this contribution whereas none of the analysed samples contained the major aflatoxin - AFB1, and the occurrence of other mycotoxins mentioned above is not so typical of meat products, as proven not only by other authors, but also by this research group during our earlier studies on dry-cured sausages (Kudumija et al., 2020. Food Addit. & Contam. B, 13, 225-232). In light of the foregoing, we were of the opinion that the results pertaining to other mycotoxins less relevant within this context, needn’t be presented just because they exist as such.
COMMENT: I think authors should mention the time frame for the sample collection.
ANSWER: We are most obliged to the Reviewer for a very useful comment. The revised Materials and Methods section now informs not only of the fact that the sampling was carried out over a two-year period (2020-2021), but also of the fact that it took place on three separate occasions (in spring, summer, and autumn), depending on the maturation of the targeted products. It is now also mentioned that the ripening time depends on the product type (for example, from 2 months for some dry-fermented sausages to 17 months for some prosciuttos).
COMMENT: What are the current legislation limits of these mycotoxins in food products?
ANSWER: European Union, as well as Croatian legislation, has still not stipulated the maximal limits (MLs) for mycotoxins in meat and meat products (Commission Regulation (EC) No 1881/2006). The above does not apply to some European countries that have defined OTA MLs in offal and meat. The countries in question are Denmark (10 µg/kg in pig kidneys), Estonia (10 µg/kg in pig liver), Romania (5 µg/kg in pig kidneys, liver, and meat), and Slovakia (5 µg/kg in meat and milk), while Italy recommended that OTA values in pork meat and pork meat products should not surpass 1 µg/kg. European Commission (EC) regulations have defined OTA and AFB1 MLs in various foodstuffs, in specific, OTA MLs in cereals and cereal-based products, dried fruit, coffee, wine, grape juice, cereal-based processed food, food intended for infants, babies, and medicinal diets, spices, and liquorice, and AFB1 MLs in tree nuts, peanuts, dried fruits, cereals, corn, milk, baby food, and spices. Pursuant to the Commission Regulation (EC) No 1881/2006, AFB1 ML in spices likely to be used in meat products’ production (chilli, paprika, white & black pepper, nutmeg, ginger, and curcuma) equals to 5 µg/kg, while that for OTA equals to 15 µg/kg, or 20 µg/kg in case of chilli and paprika.
COMMENT: I recommend presenting the data in a form of table indicating the type of mycotoxin detected and its amount in different types of samples. For example, how many samples of domestic sausages, cured hams, bacon, dry rack were collected and what is the level of mycotoxin detected in it?
ANSWER: We are grateful to the Reviewer for this most constructive comment that made the revised presentation of the study results even more detailed. Data are now also presented in form of a table informative of each type of meat products under study. To avoid data abundance and aid to the contribution clarity, the table referred to above is enclosed in form of the supplementary material “S1”.
COMMENT: Is the contamination occurring during the processing of meat preservation and packaging or is it present in the animal body, which may also come from the feed they were given?
ANSWER: Mycotoxin contamination of TMPs can occur through three pathways: consequent to the presence of mould producers that overgrow the surface of these products during meat processing, due to contaminated spices used according to the recipe, or through contaminated feed and carry-over of mycotoxins into raw materials used in the production. The last two pathways have mostly been investigated for OTA, and less thoroughly for other mycotoxins under this study. Pigs have been shown to be very sensitive to OTA transfer from contaminated feed, so that this mycotoxin is mostly found in pork meat products made from offal (Perši et al., 2014. Meat Sci. 96, 203–210). The occurrence of CIT, STC, and CPA in meat products has been underexplored even on a global scale, so that the pathways of TMP contamination with these mycotoxins are yet unknown. CPA, which was detected in the highest percentage of our study samples, even in very high concentrations, has been proven to be stable and prone to accumulation in meat products, but the studies dealing with the subject-matter are still few in numbers. Nevertheless, the presence of CPA is mostly attributed to its production by toxicogenic moulds growing on TMP surfaces, especially to P. commune. This issue definitely calls for further research, as clearly stated by the sentence added into the revised Conclusion section of this contribution, stating that “the obtained results indicate the necessity of further research into all potential sources and mechanisms of contamination during all stages of TMP production and storage, potentially responsible for the occurrence of mycotoxins in these types of products” (the addition in question being also in line with the comment of the Reviewer 1).
COMMENT: I strongly recommend improving the introduction and discussion section of the manuscript. Write how these mycotoxins are incorporated in the food chain and being toxic to the living organisms present in the environment. The below mentioned papers are suitable for citation: Deepshikha et al., 2022. Front Nutr. 9:851787; Dey et al. 2021. Environ Pollut. 268.115713; Dey et al. 2022. Crit Rev Food Sci Nutr. 1-22.
ANSWER: We are thankful to the Reviewer for this very useful comment, which prompted us to improve the presentation and discussion of our research results. We made every effort to extend the body text in the Introduction and the Discussion section and have supplemented our contribution and our Reference List with valuable quotations recommended by the Reviewer (references number 19-21).
The authors are indebted to our esteemed Reviewer for helpful suggestions and comments.
Round 2
Reviewer 1 Report
This paper has some value as a mycotoxin contamination survey of the TMP, but there is a lack of scientific basis for the paper as it relates to the Region of Origin and Local Weather.
If, as the author states, the results are part of a 4-year study, I recommend that the title be changed and rewritten to focus on The Occurrence of Mycotoxins in Traditional Dry-Cured Meat Products.
Author Response
Comment: This paper has some value as a mycotoxin contamination survey of the TMP, but there is a lack of scientific basis for the paper as it relates to the Region of Origin and Local Weather. If, as the author states, the results are part of a 4-year study, I recommend that the title be changed and rewritten to focus on The Occurrence of Mycotoxins in Traditional Dry-Cured Meat Products.
Answer: We area most obliged to our esteemed Reviewer for his/her valuable comment and suggestion about the title change. In view of the above, we hereby take the liberty to propose the following title: “The occurrence of five unregulated mycotoxins most important for traditional dry-cured meat products“, which we hope is in line with the Reviewer’s suggestion, but also better reflects the key contribution of our paper, stated also in the first sentence of the Key Contribution section of the original manuscript.
Reviewer 3 Report
I think authors have made changes in the current manuscript keeping reviewer's comments in mind. So, I think the manuscript is eligible for publication in the current format.
Author Response
The authors are indebted to our esteemed Reviewer for helpful suggestions and comments.